# Biomechanical Properties and Corrosion Resistance of Plasma-Sprayed Fish Scale Hydroxyapatite (FsHA) and FsHA-Doped Yttria-Stabilized Zirconia Coatings on Ti–6Al–4V Alloy for Biomedical Applications

Franklin A. Anene [1,2], Che Nor Aiza Jaafar [1,3,*], Azmah Hanim Mohamed Ariff [1,3,*], Ismail Zainol [4], Suraya Mohd Tahir [1], Bushroa Abdul Razak [5], Mohd Sapuan Salit [1,3] and Joy Anene-Amaechi [6]

1   Department of Mechanical and Manufacturing Engineering, Faculty of Engineering, Universiti Putra Malaysia, Serdang 43400, Malaysia
2   Department of Metallurgical and Materials Engineering, Faculty of Engineering, Nnamdi Azikiwe University, Awka P.M.B. 5025, Nigeria
3   Advanced Engineering Materials and Composite (AEMC), Faculty of Engineering, Universiti Putra Malaysia, Serdang 43400, Malaysia
4   Department of Chemistry, Faculty of Science and Mathematics, Universiti Pendidikan Sultan Idris, Tanjong Malim 35900, Malaysia
5   Department of Mechanical Engineering, Faculty of Engineering, Universiti Malaya, Kuala Lumpur 50603, Malaysia
6   Department of Agricultural & Bioresources Engineering, Faculty of Engineering, Nnamdi Azikiwe University, Awka P.M.B. 5025, Nigeria
*   Correspondence: cnaiza@upm.edu.my (C.N.A.J.); azmah@upm.edu.my (A.H.M.A.)

**Abstract:** Hydroxyapatite (HA) coatings on metallic implants have been extensively used in orthopedic applications to improve tissue-implant interactions, enhance their biocompatibility, and enhance their functionality. However, the expensive synthetic HA is the most widely used bioceramic for implant coatings, leading to high implants costs. Hence, this research explored the potential of an inexpensive biogenic HA derived from fish scales and FsHA/yttria-stabilized zirconia (YSZ) bioceramic coatings on a Ti–6Al–4V alloy as an alternative to synthetic HA coatings. The FsHA/YSZ powders and the coatings were examined with X-ray diffraction (XRD) and scanning electron microscopy/energy dispersive X-ray (SEM/EDX), and the surface roughness, microhardness, corrosion resistance, bioactivity, and in vitro cytotoxicity of the coatings were also determined. The morphological powder analysis revealed particles with a slightly irregular morphology and a fine spherical morphology, while the coating microstructure analysis revealed a fine lamellar morphology, with partially melted and unmelted FsHA particles, and fine microcracks along with evenly dispersed $ZrO_2$ particles. The surface roughness of the FsHA coating increased by 87.5% compared with the uncoated substrate, and the addition of YSZ significantly reduced this value. A 35.5% increase in hardness was obtained in the FsHA + 20 wt.% YSZ coating, and the FsHA coating showed a 43.2% reduction in the corrosion rate compared with the uncoated substrate; a further 73% reduction was observed with the addition of YSZ. The microstructure of the coatings after 14 days of immersion in simulated body fluid (SBF) revealed enlarged cracks and delaminated segments with well-grown apatite spherulite layers on the whole surface of the coatings, while in vitro cytotoxicity analysis showed a good cell viability of 95% at the highest concentration of the specimen.

**Keywords:** hydroxyapatite coating; surface roughness; microhardness; corrosion resistance; bioactivity; cytotoxicity

## 1. Introduction

Hydroxyapatite (HA), $[Ca_{10}((PO)_4)_6(OH)_2]$, has been used for decades as a bioactive bone substitute material, mainly as ceramics, cements, coatings and biocomposites in

biomedical applications [1–3]. HA has a similar chemical composition to the mineral component of human bones and hard tissues in mammals, and its excellent biocompatibility and osteoconductivity with human body fluid favors early bonding between bone tissues and implant surfaces [4,5]. Despite the excellent properties of HA, its poor mechanical properties (such as its low tensile strength, brittleness, fretting fatigue, toughness, impact resistance, and adhesive strength) have limited its use in some load-bearing applications [6].

Reinforcing HA with ceramics or metal powders has been reported to enhance its biomechanical properties. Si and $SiO_2$ [7,8], yttria-stabilized zirconia (YSZ) [9], Zn [10], and Zr [11] are some of the elements added to HA to enhance its bioactivity and mechanical properties. Zr has been widely used as a biomaterial in implant and prosthesis productions due to its biocompatibility and good mechanical strength [12,13]. Similar to Zr, reinforcement with YSZ is often used due to its high strength, biocompatibility and toughening properties during crack-particle interactions [14]. Improved mechanical properties have been reported with plasma-sprayed HA/YSZ coatings compared with bulk HA alone [15,16]. This is because of the formation of a non-transformable tetragonal (t′) phase during plasma spraying that strengthens the brittle bulk HA [17]. Additionally, Duong et al. reported that carbon nanotube (CNT) fibers with their aligned CNT structures and excellent mechanical properties are promising materials that can be used to fabricate high-performance, lightweight, and multifunctional CNT/HA composites, whereas Tebyanian et al. reported enhanced bone regeneration and new bone growth at 4 and 8 weeks after the implantation of a collagen/β-tricalcium phosphate bone graft [18,19].

Titanium and its alloys are extensively used in implant production for load-bearing applications due to their good biocompatibility and mechanical properties. However, they exhibit poor osteoconductivity, and many surface treatments have been proposed to enhance their biological properties. The use of a hydroxyapatite (HA) coating on Ti alloy implants is one of the most developed and efficient surface treatments that can optimize the mechanical properties of Ti with the bioactivity of HA [20]. Implants are expected to remain intact in the human body for an appreciable number of years (15–20 years). Unfortunately, corrosive body fluids such as blood and constituents of body fluids such as water, chlorine, sodium, proteins, plasma, and amino acids can adversely affect the biocompatibility and mechanical integrity of implants. Hence, an implant should possess a high corrosion resistance to limit its surface oxide films' dissolution that introduces toxic ions to the body, thus inducing implant failure [21,22]. The plasma spray technique is the most commercially adopted technique for HA coatings due to its high uniform deposition rates, simplicity, and low substrate temperature. Plasma coating involves the production of an ionized gas (plasma) in which HA powder is injected, partially melted, and projected to stick on a substrate at a controlled distance from the spaying gun [23–25].

Synthetic HA is stoichiometric and basically composed of calcium and phosphorus, with a Ca/P molar ratio of 1.67, and it has been reported to be most effective in promoting bone regeneration [26]. On the other hand, natural HA is non-stoichiometric due to the presence of trace elements such as Ba, Si, F, Zn, Mg, K, Na, and $CO_3$, which makes it similar in chemical composition to human bone [26,27]. HA extracted from fish scales using the calcination method has a Ca/P ratio in the range of 1.62–1.71, which is close to the stoichiometric synthetic HA Ca/P ratio of 1.67, and contains trace elements such as Na, Sr, Mg, and K [28]. The presence of these trace elements in natural HA mimics the apatite produced by human bone and plays a vital role in the regeneration of bone, as well as accelerating the process of bone formation [29,30]. Magnesium ions act as micronutrients for metabolic activities in tissues [31], and their deficiency in bone affects bone growth, skeletal metabolism, the generation of osteopenia, and bone fragility, as well as reducing osteoclastic and osteoblastic activities [32]. Similarly, zinc is an essential element in the stimulation of bone tissue formation, and it is present in bone in the range of 0.012%–0.0225%. Zn aids the prevention of localized bone inflammation by inhibiting amorphous HA's resorption [33]. Additionally, HA with traces of silicon favors cell adhesion and the development of the

organic bone phase and collagen, and 0.2wt.%–0.8 wt.% of silica has been reported to be present in biological HA [32].

Another beneficial factor in the use of natural HA derived from fish scales is that it can mitigate global environmental pollution. Recently, there has been a significant increase in fish biowaste production caused by the ever-increasing global consumption of fish. Hence, the extraction of HA from fish scales will result in a significant reduction in solid biowastes in the fishery industry and aid in combating the global environmental impact of solid biowastes. Above all, it has been reported that 40% to 50% of fish scales consist of an inorganic material known as hydroxyapatite (HA), and they have been the best alternative source of HA in biomedical applications due to low manufacturing costs and safety issues [34]. Hence, the adoption of FsHA as an implant coating material will lead to significant reductions in the cost of implants.

The present work investigated the biomechanical properties and corrosion resistance of plasma-sprayed FsHA/YSZ on a Ti–6Al–4V alloy substrate. This study was focused on the potential of using an inexpensive and natural biogenic HA derived from fish scales as an alternative coating material to synthetic HA for biomedical implants, as well as the effects of YSZ addition on the biomechanical properties of the coatings. Minimal work has been published thus far on fish scale hydroxyapatite as a coating material for biomedical implants.

## 2. Materials and Methods

### 2.1. FsHA Powder Preparation

Tilapia oreochromis niloticus fish scale (Fs) was used in this research. It is a biowaste product in Tanjung Malim market, Perak, Malaysia. The removal of impurities such as blood and salts from the Fs was conducted with soaking and washing in water, followed by air-drying. The fish scale was deproteinized by washing with 0.1 M HCl, after which it was severally washed with deionized water and dried at 60 °C in an oven. The fish scale was ground into a fine powder using a mechanical grinder. Then, 0.5 M NaOH was added to the powders, which were heated for 1 h at 100 °C. The treated fish scale powders were then thoroughly washed with distilled water until the washing solution turned neutral, and then the powders were oven-dried at 80 °C. The treated fish scale slurry was then spray-dried using a spray-drying machine. After the spray-drying, the powders were calcined at 1200 °C for 2 h to increase their crystallinity.

Next, the FsHA powders were sieved, and fine powders with particle sizes in the range of 20–50 μm were finally collected. Afterwards, fine powders of FsHA and YSZ ($ZrO_2$–8 wt.% $Y_2O_3$) purchased from Maju saintifik Sdn. Bhd. Malaysia, at compositions of 10, 15 and 20 wt.% YSZ, were mechanically mixed, ball-milled at 100 rpm for 4 h, and finally sieved after oven-drying at 100 °C. The obtained fine bioceramic powders (15–40 μm) were used for the plasma coatings after characterizations. The powders' phase composition was examined with an X-ray diffractometer (XRD, Rigaku Smartlab 3 kW, Tokyo, Japan) using CuK α-radiation at 40 kV and 40 mA with a scan rate of 0.02°/s at 10° to 70° 2θ.

### 2.2. Plasma Coating

Prior to plasma coating, the Ti–6Al–4V substrate purchased from Maju saintifik Sdn. Bhd. Malaysia, was cut into 25 mm × 10 mm × 3 mm pieces (Figure 1a) and ground in a grinding machine using silicon carbide papers of different grit sizes (180–1200). Then, it was washed for 30 min in an ultrasonicator with ethanol, acetone, and distilled water. The Ti alloys were then sand-blasted at a blasting pressure of 5 bars at a 75° blasting angle for 3 min (Figure 1b).

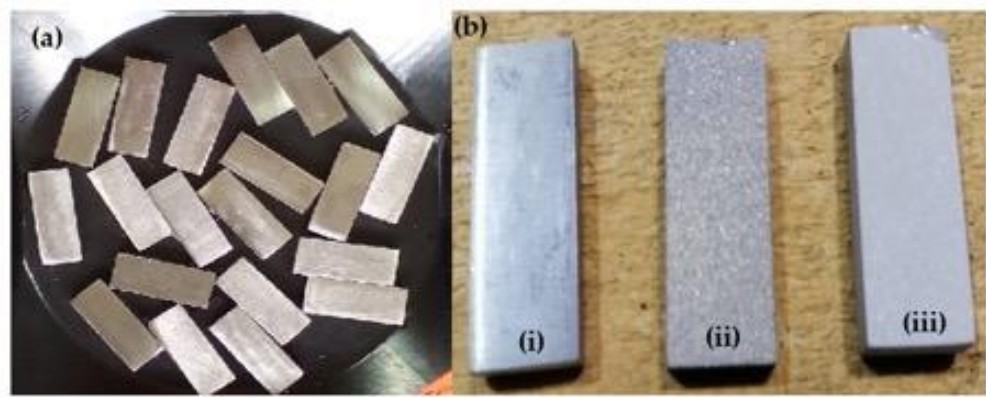

**Figure 1.** (**a**): Ti–6Al–4V alloy substrate; (**b**): (**i**) uncoated, (**ii**) grit-blasted, and (**iii**) plasma-coated.

A Praxair plasma coating machine installed in a sound-proofed, purpose-built room at the Standard and Industrial Research Institute of Malaysia (SIRIM) was used for the bioceramic depositions. A standoff distance of 100 mm was kept between the substrate and the plasma gun, and the spray process was carried out at a 250 mm/s gun transverse speed. The powder feed rate was calibrated at 15 g/min, and the carrier gas flow rate was calibrated at 30 psi (2.07 Bar). The primary plasma-forming gas was argon at a flow rate of 50 psi (3.45 Bar), and the secondary plasma-forming gas was helium at a flow rate of 40 psi (2.76 Bar). The current was maintained at 500 A.

The samples were ground after the coating using grinding and polishing machines with 320 and 1000 grit silicon carbide papers, respectively, using a 6 μm monocrystalline diamond suspension and then polished using a 0.05 μm master prep suspension. Finally, the samples were stored in a desiccator after etching for 20 s using Kroll's reagent for further analysis.

### 2.3. Surface Roughness Test

The surface roughness of the plasma-coated samples was determined using a Mahr Perthometer roughness tester (Mahr GmbH, Gottingen, Germany)). The Ra (arithmetic mean of the departures of the roughness profile from the mean line), Rz (average of the highest peaks and lowest valleys) and Rmax (average of the highest peaks) parameters were measured three times at different regions on the coated surfaces, and their mean values were calculated.

### 2.4. Microhardness Test

The hardness of the coatings was determined with a digital micro Vickers hardness tester (Mitutoyo, Aurora Illinois, CO, USA)) with a 300 gf load at 10 s of dwell time. After the indentation, measurements of the D1 and D2, diagonal length of the diamond pyramid were taken, and the hardness values that were automatically given by the machine were recorded. Each sample was tested three times at different sections with a gap of at least 1 mm, and the mean values were calculated.

### 2.5. Electrochemical Corrosion Test

Potentiodynamic polarization tests were performed in order to examine the corrosion behavior of the coatings. The tests were conducted using a Gamry instrument (Potentiostat/galvanostat/ZRA, Gamry instruments, Warminster, PA, USA)) with the electrochemical DC 105 software. A phosphate-buffered saline (PBS, Gibco by Life Technologies Corporation) solution with pH 7.4 served as the electrolyte to simulate body fluids. The coated samples were immersed in the electrolyte for 3 h at a temperature of 37 °C prior to the corrosion test for stabilization, and 1 cm$^2$ of each sample was exposed to the electrolyte during the test.

The as-coated alloys served as the working electrodes, the reference electrode was a standard Ag/AgCl one, and the counter electrode was a graphite rod. The test was carried out at a 1 mV/s scan rate for 1 h, with fresh electrolyte used for each test. Additionally, the polarization curves were initiated from −250 mV to +250 mV relative to the open circuit potential (OCP). The Ti–6Al–4V alloy was found to have an equivalent weight of 11.90, as calculated using Equation (1) and Table 1 [35].

$$Equivalent\ weight\ (EW) = \frac{1}{\sum \frac{NiFi}{Mi}}$$
$$EW = 1/(0.0747 + 0.00667 + 0.00265) = \frac{1}{0.08402} = 11.90$$

(1)

where Mi = atomic weight of the elements in the alloy; Fi = mass fraction of the elements in the alloy; and Ni = valency of the element in the alloy.

**Table 1.** Ti–6Al–4V alloy equivalent weight calculation.

| Ti–6Al–4V | | | | |
|---|---|---|---|---|
| **Element** | **Fi** | **Ni** | **Mi (g/mol)** | **NiFi/Mi** |
| Ti | 0.895 | 4 | 47.90 | 0.0747 |
| Al | 0.060 | 3 | 26.98 | 0.00667 |
| V | 0.045 | 3 | 50.94 | 0.00265 |

*2.6. In Vitro Bioactivity Assessment*

It has been established that the formation of a bone-like apatite layer on the surface of an artificial implant is one of the major requirements for the implant to bond well to the living bone and tissue [36]. The bioactivity, used to evaluate the apatite-forming ability of the FsHA/YSZ coatings, was examined in the SBF solution. The used SBF, with similar ion concentration to that of human blood plasma, was procured from Fisher Scientific International, as shown in Table 2. The samples were immersed in 20 mL of the SBF solution in plastic containers and placed in an oven maintained at 37 °C. The SBF solution was refreshed daily throughout the duration of the test to maintain a pH value of 7.4. After immersion for 14 days, the samples were removed from the oven, rinsed with distilled water, and then air-dried. Finally, the samples were stored in a desiccator for further characterization with XRD and SEM-EDX to observe the changes in their phases and morphologies. This study was carried out in strict adherence to the ASTM F1926/F1926M Standard Test Method [37]

**Table 2.** Nominal ion concentration of SBF in comparison with human blood plasma.

| Ion | Ion Concentration (mM) | |
|---|---|---|
| | **Blood Plasma** | **SBF** |
| $Na^+$ | 142.0 | 142.0 |
| $K^+$ | 5.0 | 5.0 |
| $Ca^{2+}$ | 2.5 | 2.5 |
| $Mg^{2+}$ | 1.5 | 1.5 |
| $Cl^-$ | 103.0 | 147.8 |
| $HCO_3^-$ | 27.0 | 4.2 |
| $HPO_4^{2-}$ | 1.0 | 1.0 |
| $SO_4^{2-}$ | 0.5 | 0.5 |
| pH | 7.2–7.4 | 7.4 |

*2.7. In Vitro Cytotoxicity Test of Plasma-Sprayed FsHA/YSZ Coating*

The assessment of medical devices follows a three-step method: material characterization, toxicological risk assessment, and biological testing. In all, in vivo testing is essential, but recent advances in technology have shifted away from traditional animal toxicity models to cell-based assays that identify the actual mechanisms of the chemical activity in cells [38]. To evaluate the cytotoxicity of the FsHA/YSZ coatings, the cell viability percentage was determined. Plasma-coated FsHA + 20 wt.% YSZ was chosen for this test, which was carried out in the Advanced Materials Testing Laboratory, SIRIM Industrial Research, Selangor, Malaysia.

The cell line used for the cytotoxicity test was the L929 mouse subcutaneous connective tissue fibroblast cell line (Mus musculus, NCTC clone 929, CCL-1) cultured in a standard Dulbecco Modified Eagle's Medium (DMEM).

Test Procedure and Cell Viability Evaluation

This test utilized rapid analysis with colorimetric methods and numerical assessment against the subjective visual morphological scoring method in line with the International Standardization organization ISO 10993-5 [39] recommendation of quantitative cytotoxicity assessment over qualitative methods. The test has demonstrated its usefulness in human risk assessment, which is the major objective of the ISO 10993 standard test. The following sequence of operations was carried out in accordance with the ISO 10993-5(E) standard.

1.  Extracts for the cell viability test were obtained from the coated surface of each sample after immersion in complete media for 24 h at 37 °C without agitation, with a weight-volume ratio of 200 mg/mL.
2.  The extraction vehicle (complete media) represented the negative control with no material.
3.  The pure extracts were then diluted with the complete media to make weight-vol-ume ratios of 100, 50 and 25 mg/mL. Subsequently, the pure extracts and the diluted extracts were added to a healthy monolayer of L929 cells (which were seeded with $3 \times 10^5$ cells/mL in 24-multiwell plates for 24 h) and incubated in a $CO_2$ incubator (Bioevopeak, Shandong, China) at 37 °C/5% $CO_2$ for 24 h.
4.  At the end of the 24 h incubation period, cell viability was tested using an Alamar Blue assay. The culture was stained with an Alamar Blue solution (1:10) and incubated for 4 h at 37 °C in a $CO_2$ incubator.
5.  After the 4 h incubation, the stained culture was detected with absorbance using a Universal Microplate Reader at 570 nm.

## 3. Results and Discussion

*3.1. XRD Characterization of the FsHA/YSZ Powders*

The XRD pattern of the FsHA powder is shown in Figure 2, with the Miller indices labelled, while Figure 3 shows the JCPDS No. 00-009-0432 matched with the FsHA powder peaks. The highest peak intensity of the FsHA powder was located at 31.8° 2θ°, representing the (211) crystal plane that is the crystalline HA peak according to the JCPDS for calcium phosphate materials, and the amorphous phases were located between $35^0$ and 55° 2θ as a broad hump. Figure 4 shows the XRD patterns of the FsHA coatings with 0, 10, 15, and 20 wt.% YSZ powders. The XRD spectra revealed that the YSZ addition to FsHA powder introduced major YSZ peaks at 30.17°, 50.22°, and 60.04° 2θ° for the FsHA/YSZ powders. Increases in the YSZ content increased the intensity of the YSZ peaks, as the FsHA + 20 wt.% YSZ powder had the highest YSZ peak intensity compared with the other compositions. Correspondingly, the highest peak intensities of the FsHA/YSZ powders were located at 31.8° 2θ°, representing the crystalline HA with the (211) crystal plane, and the amorphous phases were also located between 35° and 55° 2θ.

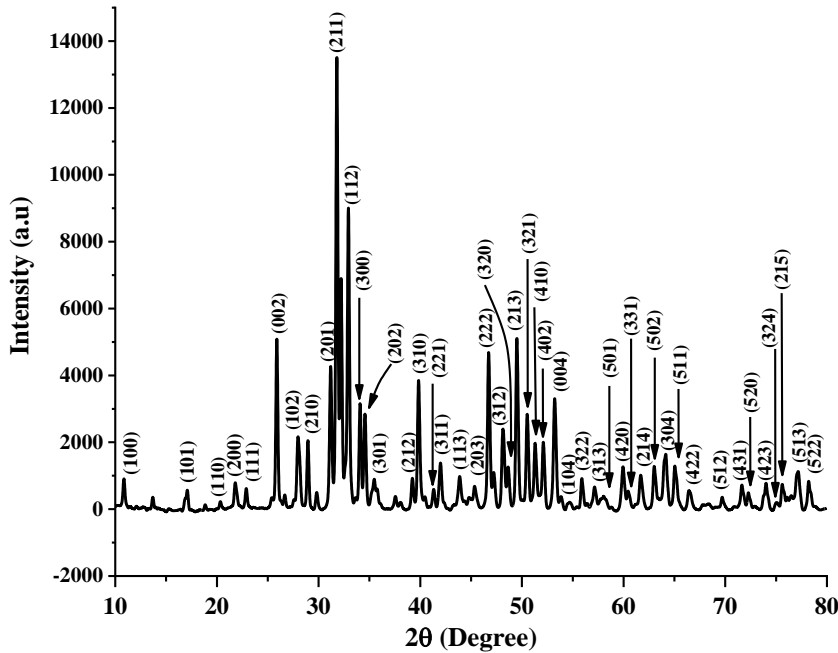

**Figure 2.** XRD pattern of FsHA powder peaks labelled with Miller indices.

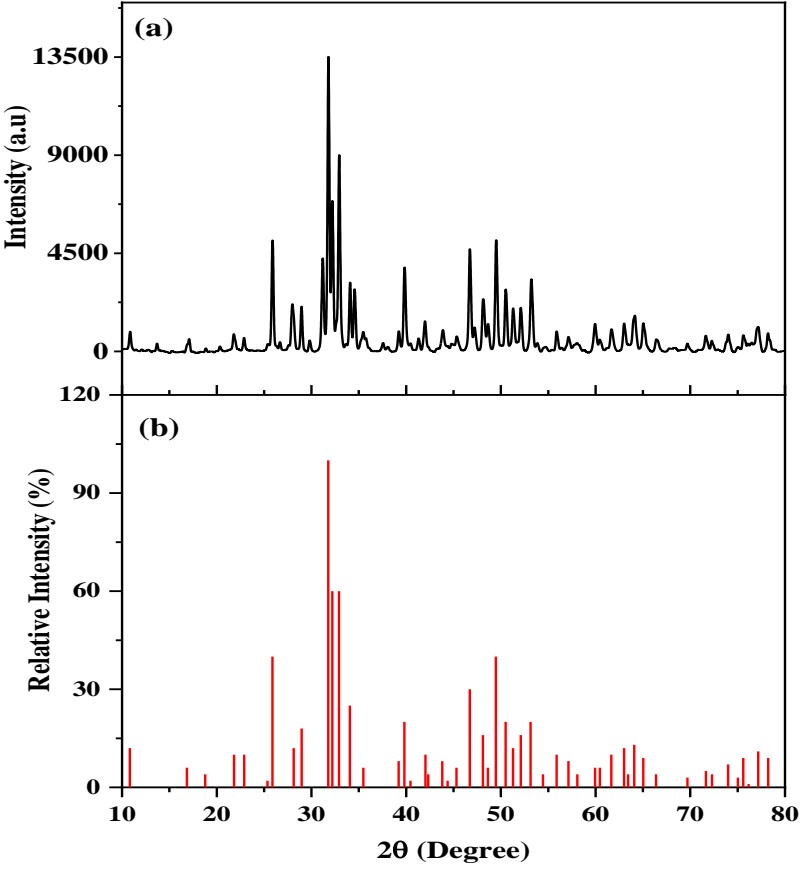

**Figure 3.** XRD patterns of (**a**) FsHA powder and (**b**) JCPDS No. 00-009-0432.

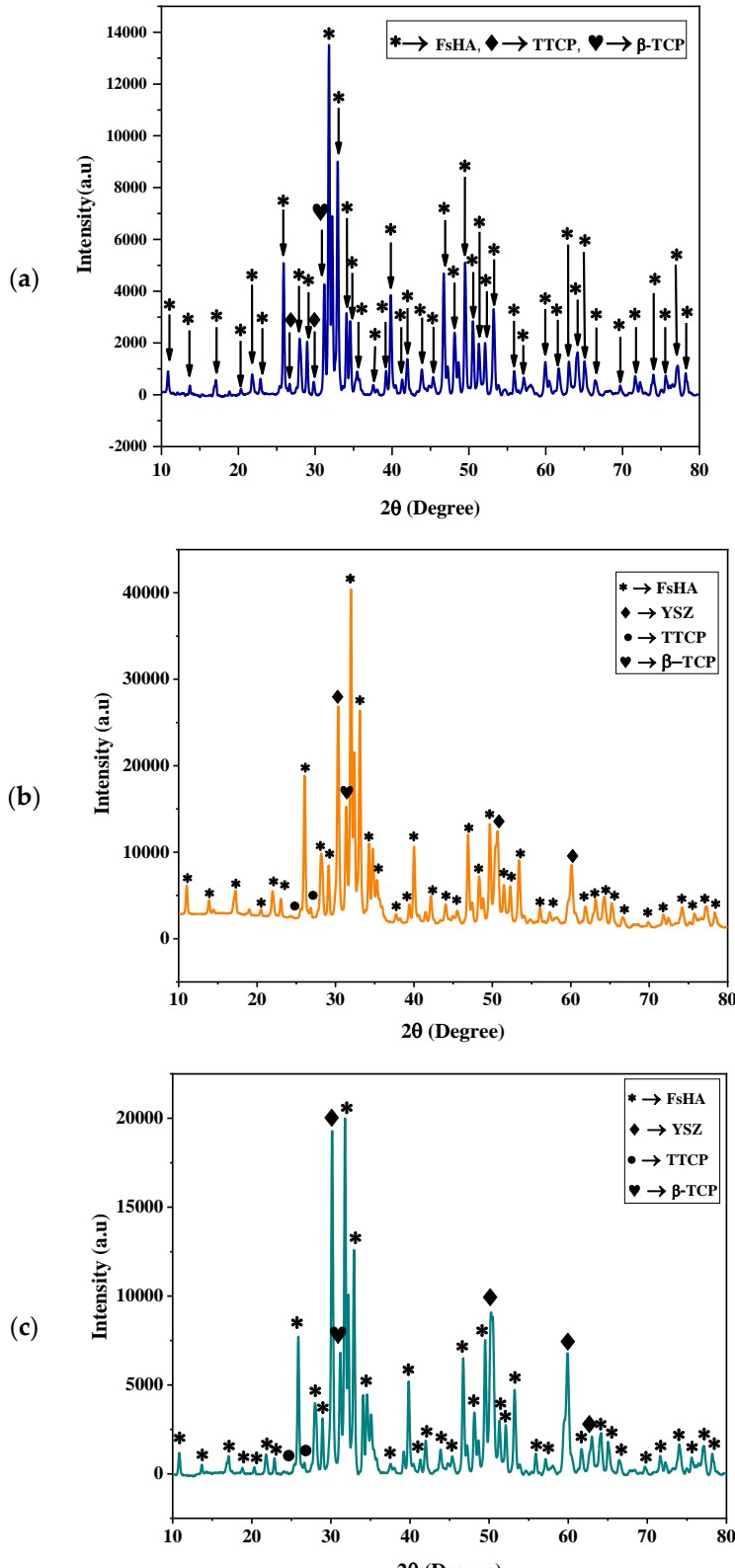

**Figure 4.** *Cont.*

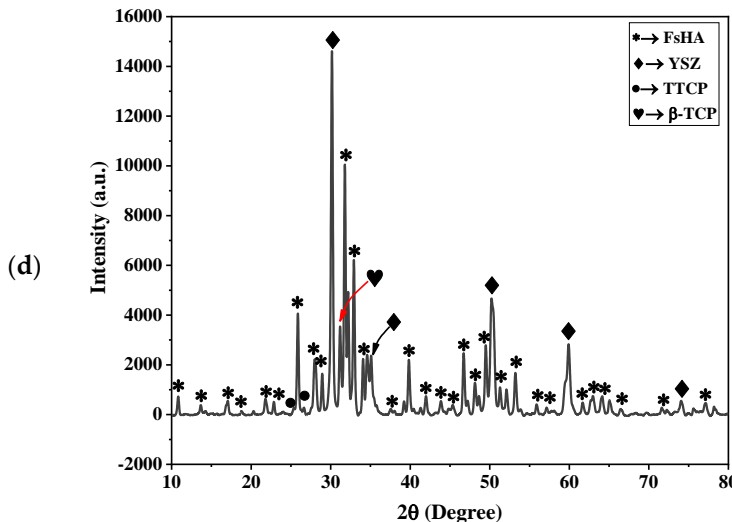

**Figure 4.** XRD patterns of FsHA powders with (**a**) 0 wt.% YSZ, (**b**) 10 wt.% YSZ, (**c**) 15 wt.% YSZ, and (**d**) 20 wt.% YSZ.

The XRD patterns revealed that crystallized FsHA/YSZ powders were obtained after the spray-dry, calcination, and ball-milling process, and they consisted of cubic/tetragonal zirconia with crystalline HA powders. The crystallinity of the FsHA/YSZ powders was determined from their respective XRD spectra using Equation (2). The percent crystallinity of the FsHA powder was 99.3%, which exceeded the 95% crystallinity required for the HA powders used in medical applications in accordance with ISO 13779-2008 [40]. Consequently, the crystallinity of the FsHA powders with 10, 15, and 20 wt.% YSZ was 98.08%, 97.37% and 96.75%, respectively. A slight decrease in the crystallinity was observed with an increase in wt.% YSZ. However, all the powders met the 95% crystallinity criterion for medical applications.

$$Crystallinity = \frac{Area\ of\ crystalline\ peaks}{Area\ of\ all\ peaks\ (crystalline + amorphous)} \times 100 \qquad (2)$$

*3.2. Microstructure Analysis*

Figure 5 shows the SEM microstructure of the FsHA + 0 wt.% YSZ and FsHA + 20 wt.% YSZ powders. The micrographs reveal similar morphologies consisting of particles with a slightly irregular morphology and particles with a fine spherical morphology, which are characteristic of spray-dried powders. In addition, the FsHA + 20 wt.% YSZ powder showed more agglomerated powder particles due to the ball-milling process. This is good for plasma spraying because particles with large amounts of irregularly shaped particles are unevenly heated by a plasma flame, resulting in compromised coating properties. Additionally, powder particles with highly irregular morphologies have been reported to have poor particle flowability in hoppers and powder feed hoses, as well as flow instability during the spraying process [41]. According to the micrographs, the fraction of the irregularly shaped particles present in the powders was small, with a lesser degree of irregularity. Thus, the powder was ideal for this work. The EDX analysis of each powder indicated the strong existence of Ca and P elements, as well as Mg, Na, and Si as trace elements for both powders. The powders had a Ca/P ratio of 1.81 for the FsHA powder and a Ca/P ratio of 2.32 for FsHA + 20 wt.% YSZ powder, which also showed traces of Zr.

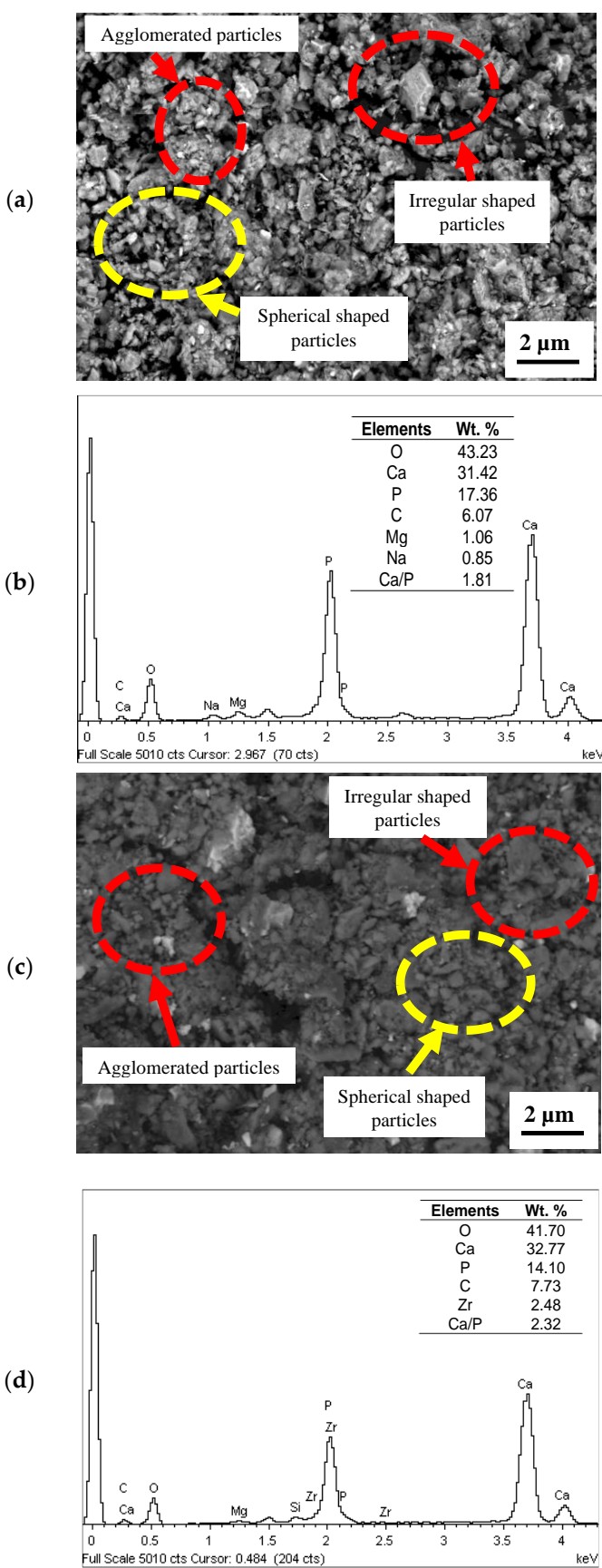

**Figure 5.** SEM micrographs of (**a**) FsHA + 0 wt.% YSZ powder, (**b**) EDX of FsHA powder, and (**c**) FsHA + 20 wt.% YSZ powder. (**d**) EDX of FsHA + 20 wt.% YSZ.

Figure 6 shows the SEM microstructure of the plasma-sprayed FsHA/YSZ coatings. Microcracks and pores with melted and unmelted spheroidized FsHA particles characterized the FsHA + 0 wt.% YSZ-coated surface (Figure 6a). Four different spectra on the micrograph (Figure 7a) were analyzed with EDX to identify the present elements, and all spectra showed strong evidence of the Ca and P elements as the main constituents. Additionally, the FsHA + 10 wt.% YSZ coating demonstrated lesser pores compared with the FsHA + 0 wt.% YSZ coating, as well as melted and unmelted YSZ and FsHA particles with minor cracks (Figure 6b). The EDX analysis of the FsHA + 10 wt.% YSZ powder (Figure 7b) indicated an adequate amount of FsHA, with 35.75 wt.% of Ca and 13.02 wt.% of P in spectrum 1, while spectra 2 and 3 consisted of gray areas of Ca and P and white dots of $ZrO_2$, seen as a solid solution of FsHA and YSZ particles that favored mechanical properties and corrosion resistance. Spectrum 4 revealed that the evenly dispersed inter-lamellar white dots within the FsHA matrix were YSZ particles. In Figure 6c, the FsHA + 15 wt.% YSZ coating micrograph reveals a dense coating, minor cracks, a smaller number of pores, and a reduced size of unmelted FsHA particles compared with the FsHA coating, in addition to partially melted YSZ particles that were evenly dispersed within the FsHA matrix. The EDX analysis of the spectra (Figure 7c) indicated an average of 33.44 wt.% of Ca, 12.81 wt.% of P, and 41.75 wt.% of O present in the coating.

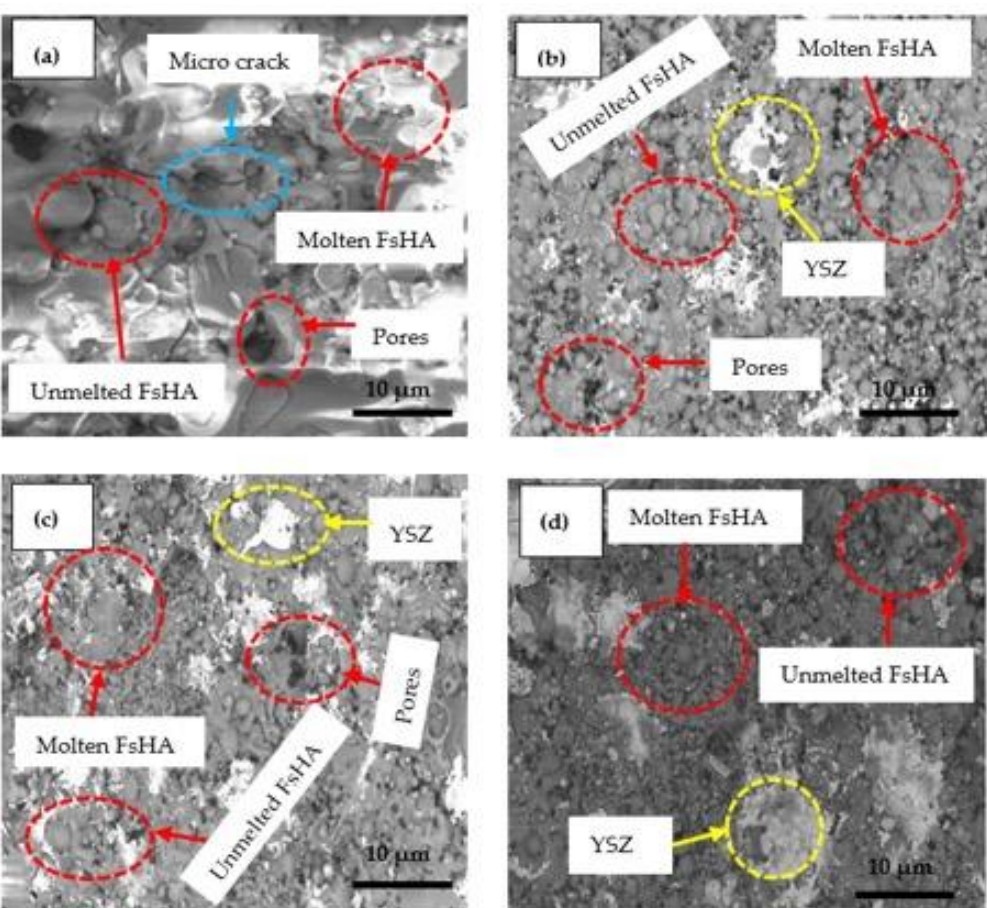

**Figure 6.** SEM micrographs of FsHA with (**a**) 0 wt.% YSZ, (**b**) 10 wt.% YSZ, (**c**) 15 wt.% YSZ, and (**d**) 20 wt.% YSZ coatings on the Ti–6Al–4V alloy substrate.

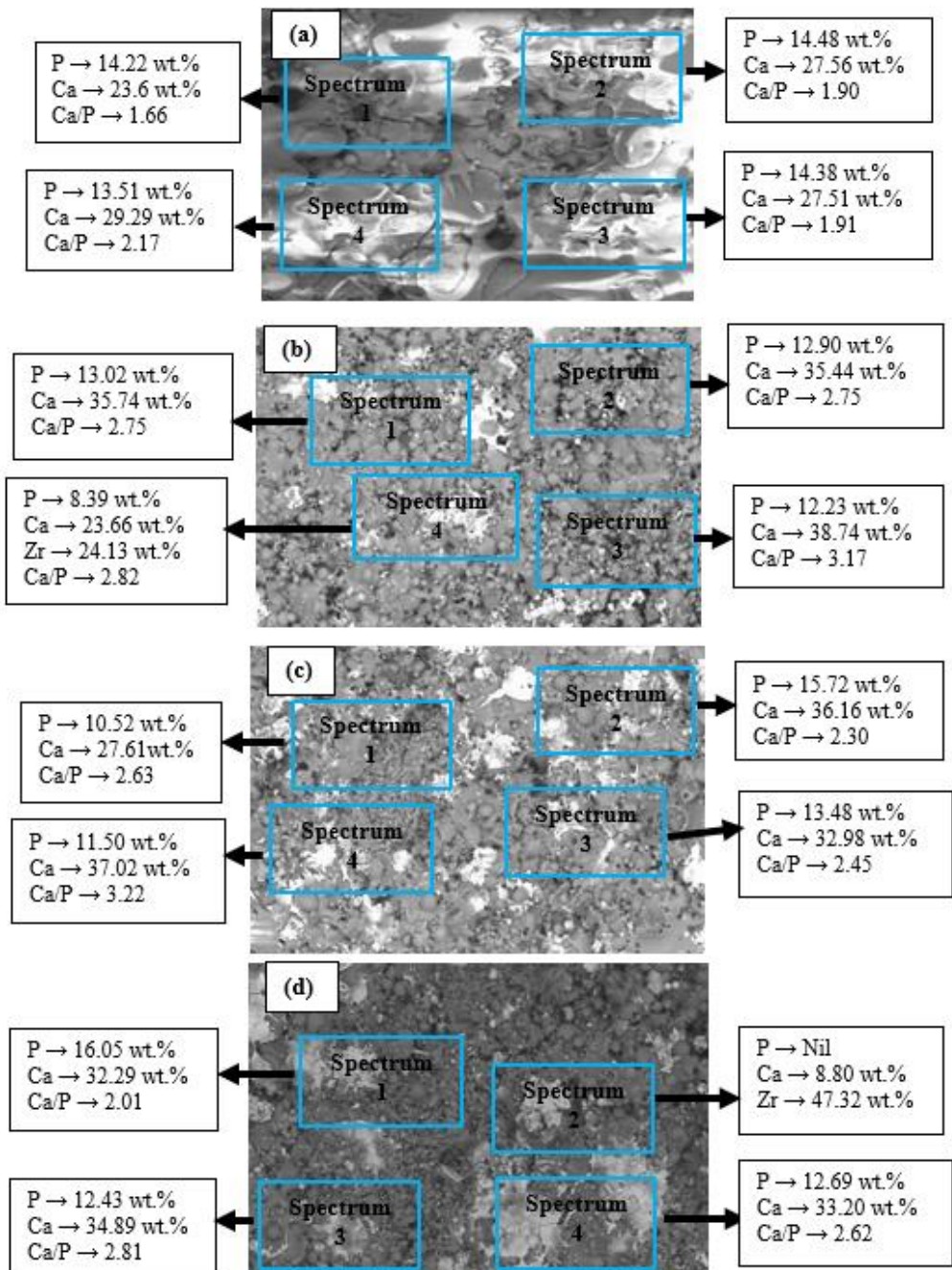

**Figure 7.** EDX analysis of FsHA with (**a**) 0 wt.% YSZ, (**b**) 10 wt.% YSZ, (**c**) 15 wt.% YSZ, and (**d**) 20 wt.% YSZ coatings on the Ti–6Al–4V substrate.

Similarly, the micrograph of the FsHA + 20 wt.% YSZ coated sample (Figure 6d) showed a higher amount of evenly dispersed white chunks of YSZ within the FsHA matrix, with partially melted and unmelted HA particles in addition to the least number of pores and the greatest density of all coated samples. Spectra 1, 3 and 4 were analyzed with EDX (Figure 7d), which revealed the strong presence of FsHA in the coatings, as all the spectra showed more than 32 wt.% of Ca and 12 wt.% of P and spectrum 2 showed 47 wt.% of Zr and 8.8 wt.% of Ca. The Ca/P molar ratio of all the coatings ranged between 1.66 and 3.22. Above all, the plasma-sprayed coatings demonstrated a fine lamellar structure with melted and unmelted FsHA particles, and the addition YSZ significantly reduced the amounts of cracks and pores and introduced evenly dispersed white chunk $ZrO_2$ particles into

the FsHA matrix, which led to a dispersion-strengthening mechanism that improved the coating's surface and mechanical properties.

### 3.3. XRD Characterization of FsHA/YSZ Coatings

The XRD pattern of the plasma-sprayed FsHA + 0 wt.% YSZ coating is shown in Figure 8. The spectrum shows crystalline peaks with the presence of an amorphous diffusion background between 37.5° 2θ° and 50° 2θ°, as well as a very high intensity peak of HA was located at 34.5° 2θ°. The diffraction pattern of the plasma-sprayed FsHA + 0 wt.% YSZ coating (Figure 8a) was found to be in line with the standard diffraction pattern of HA (JCPDS No. 00-009-0432). The very high-intensity HA peak observed in the spectrum could be attributed to the optimum coating parameters and fine powder particle size used in this work. The main phases observed in the coating were HA, CaO, TTCP, and β-TCP, which agreed with reports that plasma-sprayed HA coatings usually contain α and β-TCP, TTCP, CaO, and an amorphous calcium phosphate mixture in addition to crystalline HA [42]. Figure 8b–d also shows the XRD patterns of the plasma-sprayed FsHA coatings with 10, 15, and 20 wt.% YSZ, with the different phases identified in the coatings labelled accordingly. The XRD spectra revealed that the coatings consisted of sharp peaks of YSZ and HA and lesser peaks of the CaO, TTCP, and β-TCP phases. It was observed that the YSZ addition decreased the formation of the CaO, TTCP, and β-TCP phases, as the coating with 20 wt.% YSZ showed negligible amounts of these phases.

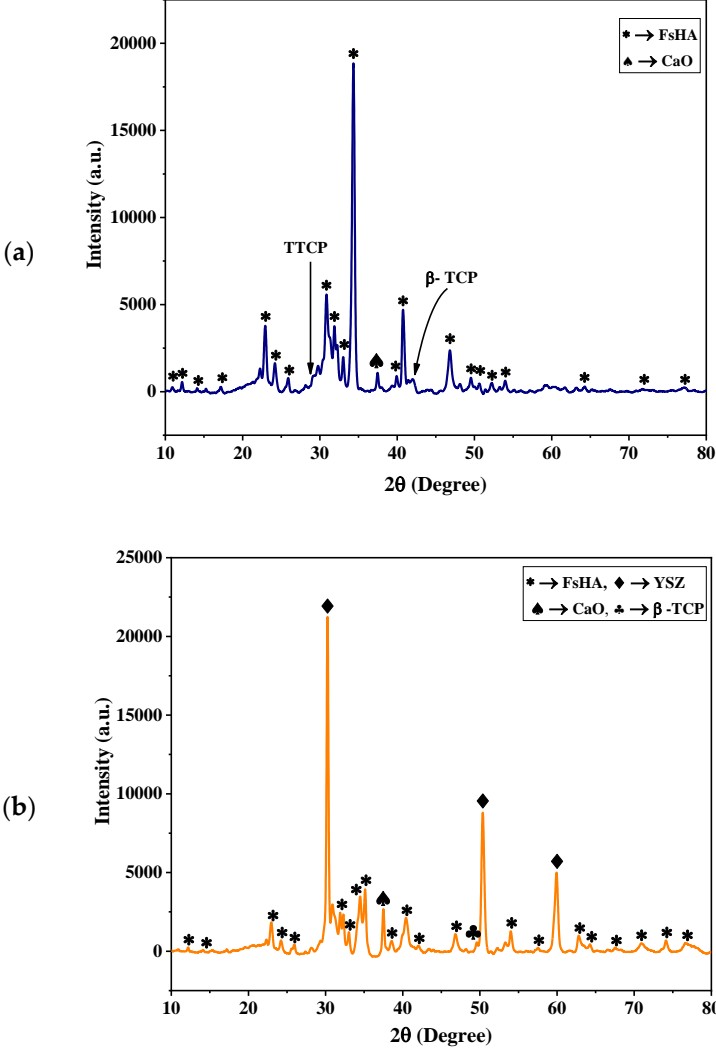

**Figure 8.** *Cont.*

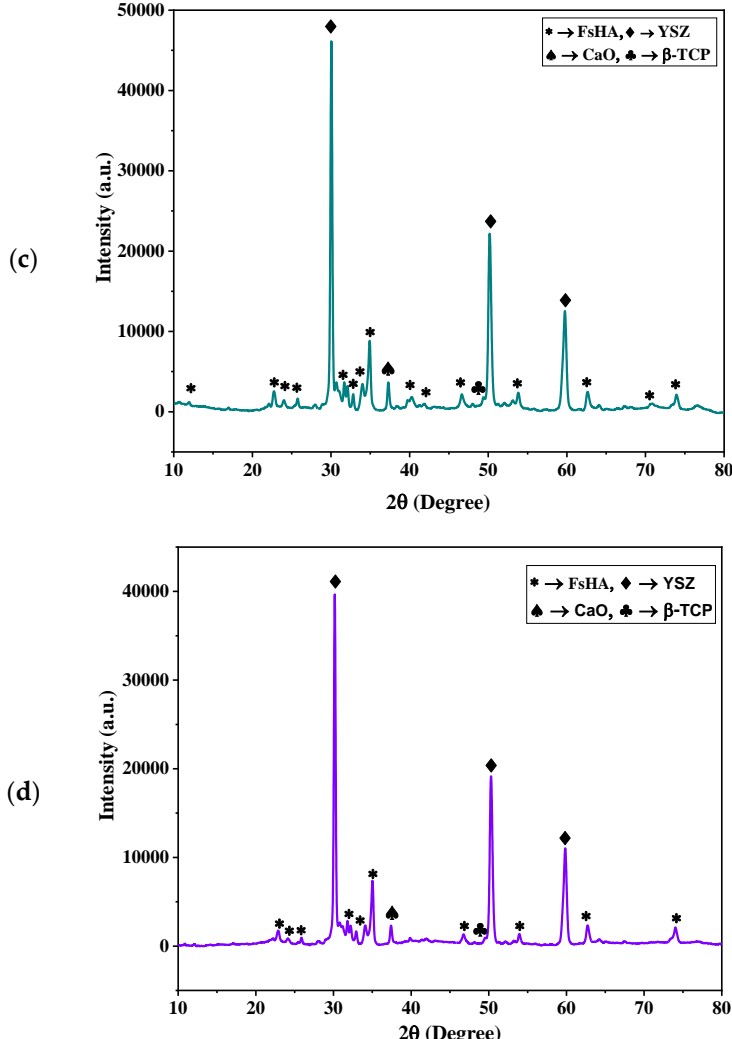

**Figure 8.** XRD patterns of plasma-sprayed FsHA with (**a**) 0 wt.% YSZ, (**b**) 10 wt.% YSZ, (**c**) 15 wt.% YSZ, and (**d**) 20 wt.% YSZ coatings.

The Crystallinity of the FsHA/YSZ coatings was less than that of their powders due to the FsHA decomposition during the plasma-spray process. The %Crystallinity of the FsHA coating was 75%, and it was observed in the XRD spectra of the FsHA/YSZ coatings that the YSZ addition led to decreased and broad HA peak intensities that resulted in slight decreases in the crystallinity of the coatings as the weight percent YSZ increased. This was attributed to the high melting point of the YSZ powder particles (2700 °C) compared with the FsHA powder particles (1670 °C), which facilitated the melting of the FsHA particles in the plasma jet. The %Crystallinity of the FsHA coatings with 10 wt.%, 15 wt.%, and 20 wt.% YSZ was found to be 68.7%, 66.54%, and 65.7%, respectively.

### 3.4. Surface Roughness

Surface roughness is described by many researchers as vital for implant–tissue interactions in clinical use. Clinically used implants are characterized by micro-pitted surfaces made by either plasma spraying, sand blasting, acid etching or laser treatment [42,43]. Table 3 lists the surface roughness values (Ra, Rz and Rmax) of the plasma-sprayed FsHA/YSZ coatings. The results showed that the least surface roughness of 0.536 μm was recorded for the uncoated substrate and that the highest roughness of 4.316 μm was recorded for the undoped FsHA coating. A slight reduction in roughness was witnessed following the addition of YSZ, and the FsHA + 10 wt.% YSZ coating's surface roughness

of 4.205 µm was the smallest of the FsHA/YSZ coatings. These results agreed with the surface roughness values of between 4 and 6.5 µm for plasma-coated HA surfaces reported by other researchers [44,45]. However, a high surface roughness increases the dissolution rate and apatite precipitation of plasma-sprayed HA coatings [46].

**Table 3.** Surface roughness and hardness of FsHA/YSZ coatings.

| Parameters | Uncoated | FsHA + 0 wt.% YSZ | FsHA + 10 wt.% YSZ | FsHA + 15 wt.% YSZ | FsHA + 20 wt.% YSZ |
|---|---|---|---|---|---|
| Ra (µm) | 0.536 | 4.316 | 4.205 | 4.252 | 4.218 |
| Rz (µm) | 3.69 | 27.1 | 28.1 | 24.50 | 28.10 |
| Rmax (µm) | 5.24 | 30.1 | 31.0 | 33.80 | 35.60 |
| Hardness (Hv 0.3) | 360.3 | 459.1 | 497.4 | 531.6 | 558.5 |

*3.5. Microhardness*

Microhardness determines the resistance of coatings to plastic deformation in service. The Vickers microhardness tests of the coatings followed the ASTM E92 standards, with the samples indented for 10 s of dwell time with a 300 gf load. The results are presented in Table 3. The hardness of the coatings showed significant improvements in comparison with the hardness of the uncoated substrate (360.3 Hv). The average hardness values of the FsHA coatings with 0 wt.%, 10 wt.%, 15 wt.%, and 20 wt.% YSZ were 459.1, 497.4, 531.6 and 558.5 Hv, respectively. The enhanced hardness of the coatings indicated that adequate bonds existed between the substrate and the coatings. The improvements in the hardness of the coatings can be attributed to the addition of YSZ to FsHA that introduced evenly dispersed $ZrO_2$ particles to the FsHA/YSZ coating, as well as increasing the amount of molten FsHA particles. These results were in line with those of other works [47,48].

*3.6. Corrosion Behavior*

For HA-coated titanium alloy implants to be accepted, the coatings must last long in vivo and possess good corrosion resistance. Figure 9 shows the potentiodynamic scans in the PBS solution of the FsHA/YSZ coatings on the Ti–6Al–4V substrate. The following corrosion parameters were determined from the potentiodynamic curves using the Tafel extrapolation method: corrosion rate (CR), corrosion current density (Icorr), corrosion potential (Ecorr), anodic Tafel slope (βa) and cathodic Tafel slope (βc). Additionally, the polarization resistance (Rp) and protection efficiency (PE) were calculated using corrosion Equations (3) and (4), respectively. The results shown in Table 4 indicated enhanced corrosion resistance for the plasma-sprayed coatings in comparison with the uncoated Ti–6Al–4V substrate. The corrosion rate of 96.137 mmpy recorded for the FsHA + 0 wt.% YSZ coating was much lower than the 169.37 mmpy recorded for the uncoated substrate and represented a 43% reduction in the corrosion rate.

$$Polarization\ resistance(Rp) = \frac{\beta a \beta c}{2.3(Icor)(\beta a + \beta c)} \tag{3}$$

where βa and βc are the anodic and cathodic Tafel constants, respectively.

$$Protection\ efficiency\ (PE) = \frac{Rp\ (coated) - Rp\ (uncoated)}{Rp\ (coated)} \times 100 \tag{4}$$

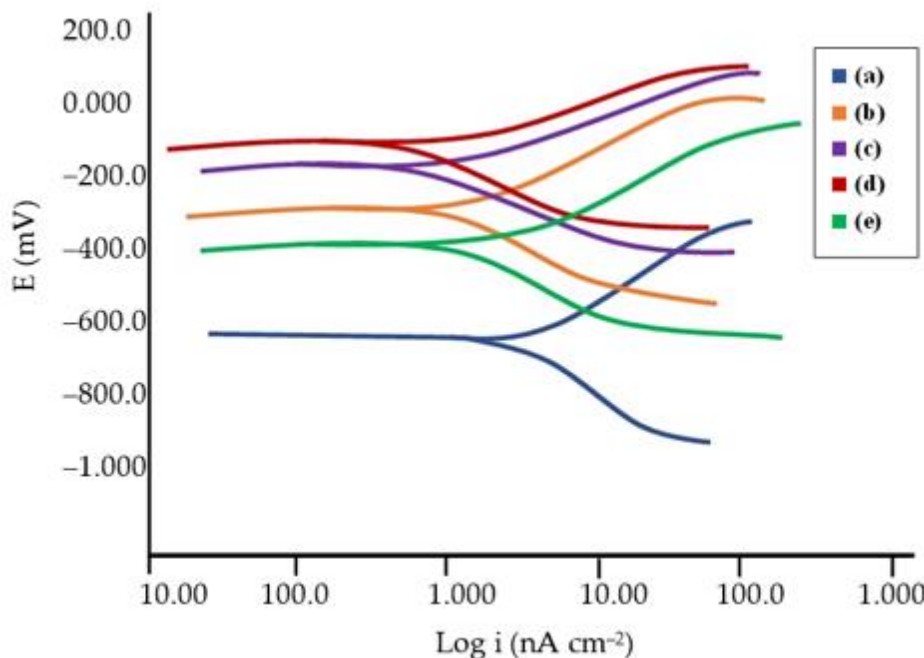

**Figure 9.** Potentiodynamic scans of (**a**) uncoated Ti–6Al–4V substrate, (**b**) FsHA + 0 wt.% YSZ coating, (**c**) FsHA + 10 wt.% YSZ coating, (**d**) FsHA + 15 wt.% YSZ coating, and (**e**) FsHA + 20 wt.% YSZ coating.

**Table 4.** Potentiodynamic test results.

| Parameters | Uncoated | FsHA+0 wt.% YSZ | FsHA + 10 wt.% YSZ | FsHA + 15 wt.% YSZ | FsHA + 20 wt.% YSZ |
|---|---|---|---|---|---|
| $\beta$a (mV/decade) | 359.1 | 160.7 | 64.8 | 82.93 | 183.2 |
| $\beta$c (mV/decade) | 202.9 | 192.6 | 58.6 | 75.76 | 70.11 |
| Ecorr (mV) | −609 | −258.9 | −143 | −87.60 | −366.3 |
| Icorr (nA cm$^{-2}$) | 485 | 275.3 | 74.3 | 36.30 | 27.11 |
| CR (mmpy) | 169.37 | 96.137 | 25.95 | 12.676 | 9.467 |
| Rp ($\Omega$ cm$^2$) | 0.1162 | 0.1384 | 0.1801 | 0.474 | 0.8132 |
| PE (%) | – | 16 | 35.5 | 75.5 | 85.7 |
| OCP (mV) | −613.5 | −233.7 | −129.3 | −55.25 | −286 |

Furthermore, a higher enhancement in corrosion rate was witnessed with the plasma-sprayed FsHA/YSZ coatings, and the FsHA with 15 wt.% and 20 wt.% YSZ coatings showed minimum corrosion rates of 12.68 and 9.467 mmpy, respectively. Similarly, the coatings had lesser corrosion current density (Icorr) values than the uncoated substrate, which indicates an enhanced corrosion resistance because the corrosion rate of a material is proportional to the corrosion current density at a given potential. FsHA coatings with 15 wt.% and 20 wt.% YSZ had the minimum Icorr values of 36.30 and 27.11 nA, respectively. Additionally, the result showed a 16% protection efficiency (PE) for the undoped FsHA coating over the uncoated substrate, and the protection efficiencies for the doped FsHA coatings with 10 wt.%, 15 wt.%, and 20 wt.% YSZ were 35.5%, 75.5% and 85.7%, respectively, compared with the uncoated substrate. These results were in line with the fact that decrease in surface roughness improves the corrosion resistance of coatings. Additionally, the evenly dispersed YSZ particles within the FsHA/YSZ coatings' structures, their lesser amount of pores, and their smaller pore sizes contributed to their enhanced corrosion resistance in comparison with the undoped coating and the uncoated substrate. These results agreed with other works that reported an improved corrosion resistance with plasma-sprayed HA and HA-doped coatings on Ti alloys [44,49,50].

### 3.7. Evaluation of In Vitro Bioactivity of Plasma-Sprayed FsHA/YSZ Coatings

One of the essential ways to determine the bioactivity of a biomaterial is to examine its apatite precipitation behavior in SBF. The formation of an apatite layer on the surface of an implant results in adequate bonding between the implant and the living bone and tissue [36,51]. Figure 10 shows the SEM microstructures of the FsHA/YSZ coatings on the Ti–6Al–4V substrate after 14 days of immersion in SBF. Enlarged cracks, fragmentation, and delaminated segments with well-formed apatite spherulite layers were observed on the whole surface of the coatings, in addition to some areas showing resistance to dissolution-especially the YSZ-rich areas (Figure 10d). The enlarged cracks were caused by stress relief during the dehydration process upon removal from the SBF solution and will not be developed in a real moist physiological environment. Additionally, cracks developed due to shrinkage upon drying have been reported by other researchers to appear in apatite layers [52].

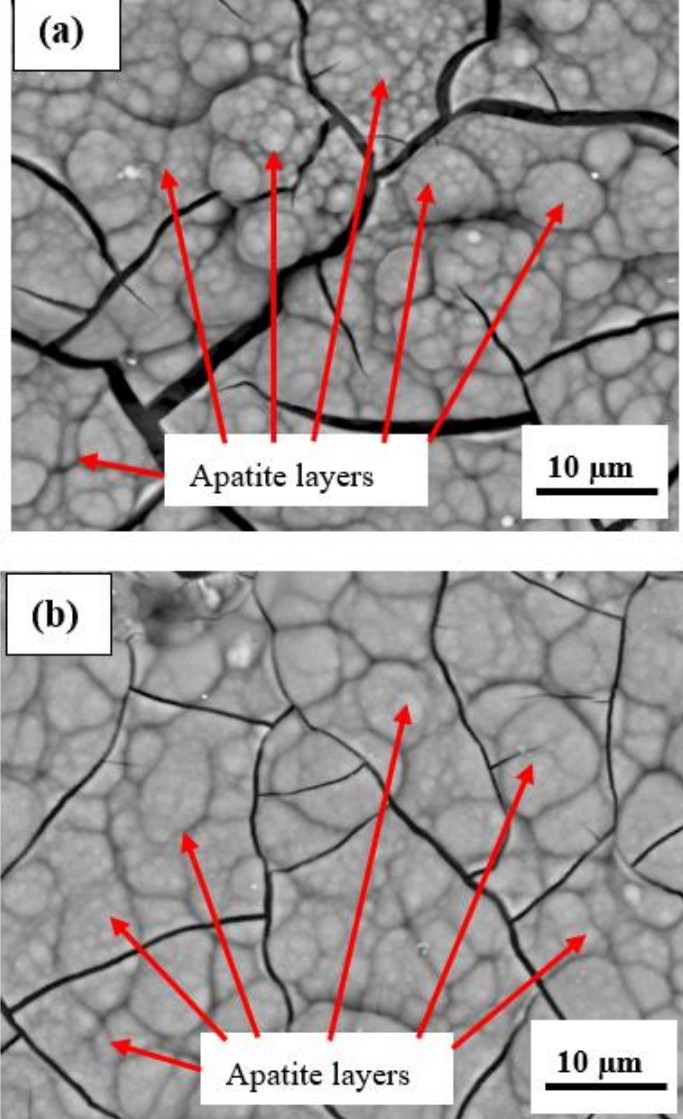

**Figure 10.** *Cont*.

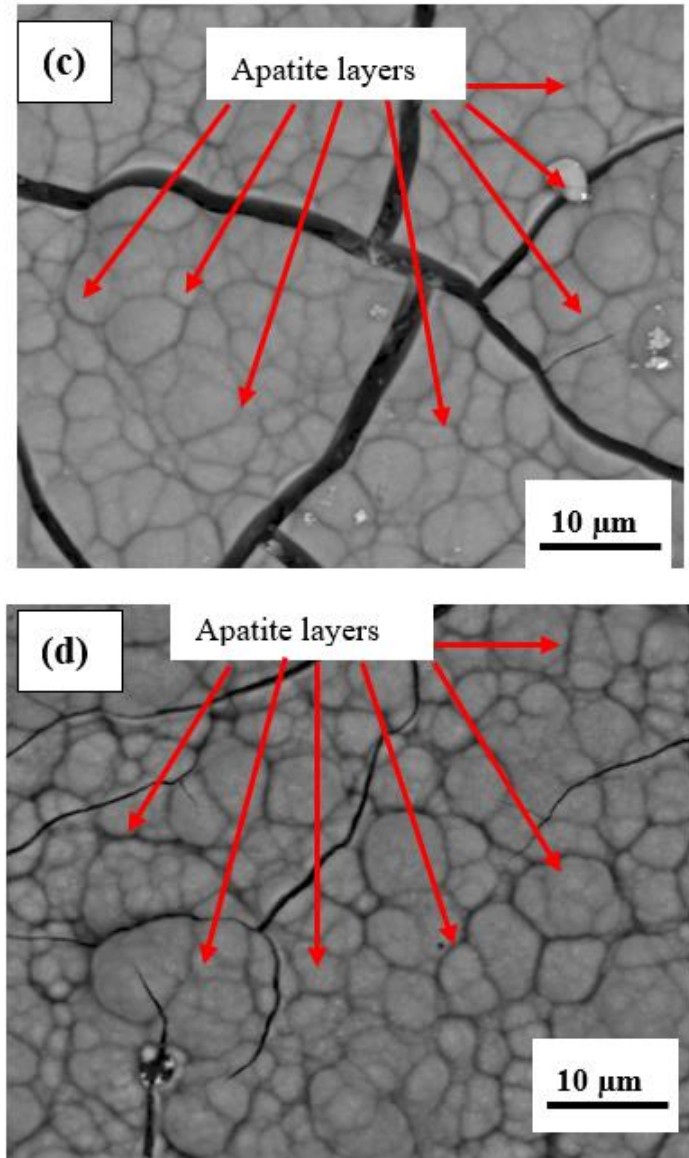

**Figure 10.** SEM micrographs of plasma-sprayed FsHA with (**a**) 0 wt.% YSZ, (**b**) 10 wt.% YSZ, (**c**) 15 wt.% YSZ, and (**d**) 20 wt.% YSZ coatings on the Ti–6Al–4V substrate after 14 days of immersion in SBF.

The observed cracks were expected, as the dissolution of the coatings in the early immersion days either exposed the fine microcracks (which already existed in the coatings) or propagated the existing surface cracks due to the release of thermal residual stresses within the coatings. These led to increases in the crack sizes and density with increases in the number of immersion days, thus indicating the dissolution process. Guo et al. [53] reported cracked regions within coatings and delaminated segments after 1 day of immersion in SBF. Additionally, the impurity phases β-TCP and CaO formed in the plasma-sprayed coatings studied here were noted to have a high resorbability and to accelerate the dissolution of coatings and the fragmentation of the lamellae morphology of the plasma-sprayed coatings.

The EDX analysis after 14 days of immersion (Figure 11) showed significant amounts of C, O, Ca, and P with Ca/P ratios in the range of 1.73–1.86, which were close to the stoichiometric ratio of 1.67. This indicated the occurrence of multiple reactions between the coatings and the SBF solution, leading to the formation of bone-like apatite $(Ca_{10-x}(PO_4)_{6-x}(CO_3)_x(OH)_2)$. The presented results clearly demonstrate that apatite layers can be developed on the surfaces of plasma-sprayed FsHA/YSZ coatings. The

ability of the FsHA/YSZ coatings to nucleate and grow apatite layers on their surfaces in a metastable SBF solution demonstrates their excellent bioactivity. Furthermore, a bioactive material's ability to develop an apatite layer is a necessary surface reaction property to form a good bond between the bioactive material and the bony tissue. Hence, it is expected that strong chemical bonds will be developed between FsHA/YSZ coatings and bone tissues after implantation.

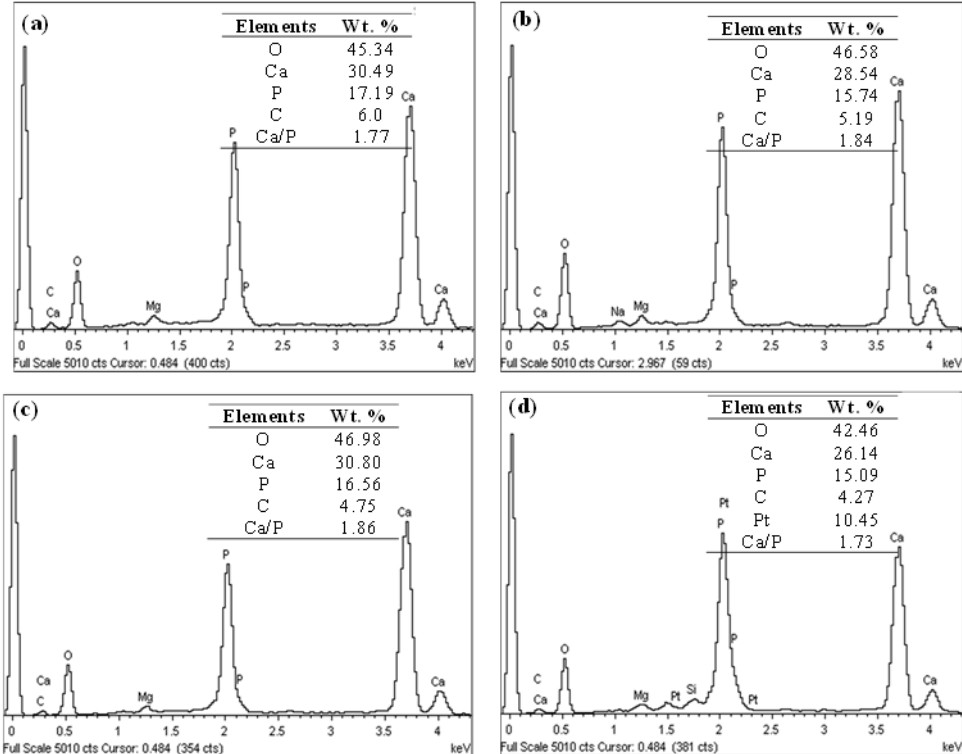

**Figure 11.** EDX analysis of plasma-sprayed FsHA with (**a**) 0 wt.% YSZ, (**b**) 10 wt.% YSZ, (**c**) 15 wt.% YSZ, and (**d**) 20 wt.% YSZ coatings on the Ti–6Al–4V substrate after 14 days of immersion in SBF.

### *3.8. Evaluation of the In Vitro Cytotoxicity of FsHA/YSZ Coating*

Toxicants have been defined as chemicals with potential adverse properties that, when released in the body, can either: (i) induce molecular, cellular or tissue/organ dysfunction by altering the local biological environment or (ii) react with particular endogenous target molecules, such as proteins, DNA, or membrane components. Perturbation(s) in normal cell function and repair processes can arise from both scenarios, thus leading to cytotoxicity [38]. The Alamar Blue Assay used in this study is a good cell viability indicator that utilizes the natural reducing power of living cells to convert resazurin to the fluorescent molecule, resorufin.

The percent cell viability was calculated using Equation (5). A low cell viability indicates that a test material would generate a cytotoxic effect. Four replicates with a growth medium (negative) and phenol 1 v% (positive) were used in this study as controls. The results showed that the L929 cells demonstrated good cell viability (95%) at the highest concentration (200 mg/mL) of test material in contrast to the positive control cultures (5%). Figure 12 shows a graphical representation of the percent cell viability at various concentrations of the coated specimen, Table 5 presents the OD and L929 cell viability values after 24 h of exposure to the coated specimen, and Table 6 lists the OD and L929 cell viability values after 24 h of exposure to the positive control. These results confirmed that the plasma-sprayed FsHA/YSZ coatings did not show any cytotoxic effects in all tested

concentrations. These results were in agreement with the ISO 10993 standard, which states that a material is considered nontoxic if its cell viability is greater than 70%.

$$\% \; Cell \; viability = \frac{\sum OD_e}{\sum OD_n} \times 100 \tag{5}$$

where ODe is the mean optical density of cells exposed with the test material and ODn is the mean optical density of unexposed cells.

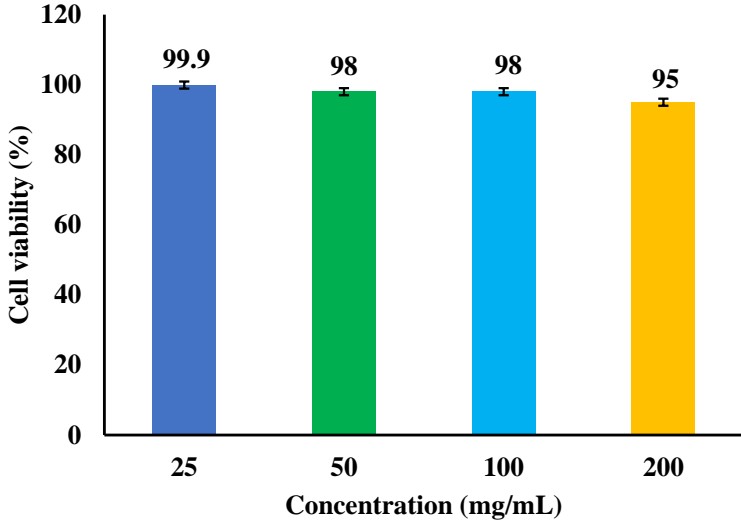

**Figure 12.** Percent cell viability at various concentration of the coated specimen.

**Table 5.** OD and L929 cell viability values after 24 h of exposure to the coated specimen.

| | Negative Control | FsHA + 20 wt.% YSZ (mg/mL) | | | |
|---|---|---|---|---|---|
| | | **25** | **50** | **100** | **200** |
| OD (570 nm) | 0.685 | 0.667 | 0.699 | 0.663 | 0.718 |
| | 0.689 | 0.753 | 0.697 | 0.714 | 0.671 |
| | 0.674 | 0.762 | 0.626 | 0.759 | 0.746 |
| | 0.855 | 0.718 | 0.829 | 0.694 | 0.629 |
| Mean (OD) | 0.726 | 0.725 | 0.713 | 0.708 | 0.691 |
| Viability (%) | 100 | 99.9 | 98 | 98 | 95 |

**Table 6.** OD and L929 cell viability values after 24 h of exposure to the positive control.

| | Negative Control | Phenol 1 v% (%) | | | |
|---|---|---|---|---|---|
| | | **12.5** | **25** | **50** | **100** |
| OD (570 nm) | 0.606 | 0.412 | 0.050 | 0.034 | 0.031 |
| | 0.566 | 0.370 | 0.048 | 0.037 | 0.031 |
| | 0.576 | 0.365 | 0.048 | 0.031 | 0.031 |
| | 0.582 | 0.336 | 0.045 | 0.036 | 0.033 |
| Mean (OD) | 0.583 | 0.371 | 0.048 | 0.035 | 0.032 |
| Viability (%) | 100 | 64 | 8 | 6 | 5 |

## 4. Conclusions

This work investigated the morphology and biomechanical properties of plasma-sprayed FsHA/YSZ coatings on a Ti–6Al–4V substrate. The main findings of the research are summarized below.

1. The XRD patterns of the powders showed the highest peak intensity at 31.8°, representing the (211) crystal plane, which is the crystalline HA peak according to the JCPDS. The Crystallinity of the powders was above 96%, and the least Crystallinity of the plasma-sprayed coatings was 65.7%. Additionally, the XRD pattern of the undoped FsHA coating consisted of a sharp peak of HA, with lesser peak intensities for the CaO, TTCP, and β-TCP phases, while that of the FsHA/YSZ coatings had an extra sharp peak intensity of YSZ.

2. The microstructures of the coatings showed significant amounts of Ca and P. The micrograph of the FsHA + 0 wt.% YSZ coating revealed micropores, microcracks, and molten and unmelted spheroidized FsHA particles, whereas the FsHA/YSZ coatings showed an increased amount of melted FsHA particles, lesser pores, fine microcracks, and $ZrO_2$ particles.

3. The highest hardness of 558.5 Hv was obtained with the FsHA + 20 wt.% YSZ coating as a result of the solid solution strengthening of YSZ in FsHA.

4. The FsHA + 10 wt.% YSZ-coated sample had the least surface roughness (4.205 μm) compared with the other coated samples: FsHA + 0 wt.% YSZ (4.316 μm), FsHA + 15 wt.% YSZ (4.252 μm), and FsHA + 20 wt.% YSZ (4.218 μm). This showed that the YSZ addition slightly reduced the roughness of the doped coatings.

5. The undoped FsHA coating showed a significantly improved corrosion resistance, with a 43% reduction in the corrosion rate compared with the uncoated substrate, and the FsHA/YSZ coatings showed further improvements in corrosion resistance, with the FsHA + 20 wt.% YSZ coating having the least corrosion rate of 9.467 mmpy.

6. The microstructure of the plasma-sprayed FsHA/YSZ coatings after 14 days of immersion in SBF revealed enlarged cracks and delaminated segments with well-grown apatite spherulite layers on the whole surface of the coatings. The EDX analysis of the coatings after 14 days of immersion in SBF confirmed the strong presence of Ca and P, with Ca/P ratios in the range of 1.73–1.86, which were close to the stoichiometric ratio of 1.67. This indicated the occurrence of multiple reactions between the coatings and the SBF solution that led to the formation of bone-like apatite $(Ca_{10-x}(PO_4)_{6-x}(CO_3)_x(OH)_2)$.

7. The in vitro cytotoxicity results showed that the L929 cells demonstrated a good cell viability of 95% at the highest concentration (200 mg/mL) of the coated specimen in contrast to the positive control cultures with a cell viability of 5%.

**Author Contributions:** Conceptualization, F.A.A., C.N.A.J. and I.Z.; methodology, F.A.A., C.N.A.J. and A.H.M.A.; validation, C.N.A.J., I.Z. and S.M.T.; formal analysis, F.A.A.; investigation, F.A.A. and I.Z.; resources, S.M.T., B.A.R. and M.S.S.; data curation, C.N.A.J.; writing—original draft preparation, F.A.A.; writing—review and editing, C.N.A.J., A.H.M.A. and I.Z.; visualization, B.A.R., J.A.-A. and S.M.T.; supervision, C.N.A.J., I.Z. and A.H.M.A.; project administration, C.N.A.J.; funding acquisition, C.N.A.J., A.H.M.A. and M.S.S. All authors have read and agreed to the published version of the manuscript.

**Funding:** This research was funded by Universiti Putra Malaysia through the Putra Grant-IPS (GP-IPS/2020/9683900).

**Institutional Review Board Statement:** Not applicable.

**Informed Consent Statement:** Not applicable.

**Data Availability Statement:** The data presented in this study are available upon request from the corresponding author.

**Acknowledgments:** The authors gratefully acknowledge Universiti Putra Malaysia for the support provided through the Strength of Materials Laboratory and the Chemistry Department, UPSI, Malaysia for support with the test analysis.

**Conflicts of Interest:** The authors have no conflict of interest in the publication of this work.

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
