# Peer review of "Biomechanical Properties and Corrosion Resistance of Plasma-Sprayed Fish Scale Hydroxyapatite (FsHA) and FsHA-Doped Yttria-Stabilized Zirconia Coatings on Ti–6Al–4V Alloy for Biomedical Applications"

_coatings, doi:10.3390/coatings13010199_

Round 1

Reviewer 1 Report

The subject of this study is very interesting and significant from a practical point of view. Namely, the authors have explored the potential of an inexpensive biogenic HA derived from fish scales and FsHA/YSZ bioceramic coatings on Ti-6Al-4V alloy as an alternative to synthetic HA coatings.

The research was extensive, well planned and executed, and the explanations in the text were understandable and logical.

With all praise for the authors, I would like to add a few small suggestions:

-        Lines 180-182: "A steady-state open circuit potential (OCV) was established during the test by conducting the test at 1 mV/sec scan rate for 1 hour while fresh electrolyte was used for each test. Also, the polarization curves were initiated at -250 mV to +250 mV relative to the OCV." This is unclear. Please correct this.

-        The abbreviation for open circuit potential is OCP.

-        Move equations (1-5) and Table 1 to the appropriate subsections in "3. Results and Discussion" with appropriate text. This will provide better context and transparency of the results, especially in the "3.6 Corrosion behaviour" section. If possible, reduce the font size in the given equations.

-       Figure 9 should be revised. The ordinate should be labeled "E (mV)" and the abscissa should be labeled "log i (nA cm-2)". In addition to the coordinates in the figure, only numerical values without associated units are required.

-        Table 4: The unit for βa and βa should be (mV/decade); the unit for Icorr should be (nA cm-2); the unit for Rp should be (Ω cm2). Also, check the values for Rp in the table (I think they should be significantly larger). Also comment on the obtained quantities for Rp.

-        Rewrite and present the basic conclusions of the paper.

Author Response

Thank you for the comments. Our response is as below:

Reviewer 1 Comments

Answers

1. Lines 180-182: "A steady-state open circuit potential (OCV) was established during the test by conducting the test at 1 mV/sec scan rate for 1 hour while fresh electrolyte was used for each test. Also, the polarization curves were initiated at -250 mV to +250 mV relative to the OCV." This is unclear. Please correct this.

The test was carried out at 1 mV/sec scan rate for 1 hour with fresh electrolyte used for each test. Also, the polarization curves were initiated at -250 mV to +250 mV relative to the open circuit potential (OCP).

2. The abbreviation for open circuit potential is OCP.

Done

3. Move equations (1-5) and Table 1 to the appropriate subsections in "3. Results and Discussion" with appropriate text. This will provide better context and transparency of the results, especially in the "3.6 Corrosion behaviour" section. If possible, reduce the font size in the given equations.

Equations 1,3,4 and 5 were moved to the appropriate subsections in 3 but equation 2 and Table 1 were not moved to subsection 3 because they were used in the methodology.

4. Figure 9 should be revised. The ordinate should be labeled "E (mV)" and the abscissa should be labeled "log i (nA cm-2)". In addition to the coordinates in the figure, only numerical values without associated units are required.

Done

5. Table 4: The unit for βa and βa should be (mV/decade); the unit for Icorr should be (nA cm-2); the unit for Rp should be (Ω cm2). Also, check the values for Rp in the table (I think they should be significantly larger). Also comment on the obtained quantities for Rp.

The units of the various parameters have been corrected. The values for Rp were as calculated from the Rp equation.

6. Rewrite and present the basic conclusions of the paper.

Done. Thank you. 

Reviewer 2 Report

Dear Editor,

This is an interesting article and authors this research explored the potentials of an inexpensive biogenic HA derived from fish scales and FsHA/YSZ bioceramic coatings on Ti-6Al-4V alloy as 25 an alternative to synthetic HA coatings. But there are some comments which should be considered before next steps:

- Abstract: aim should be clearly expressed

-Introduction needs to be improved and authors should explain the interaction between cell and mentioned materials in their manuscript and they can use the following references:  https://doi.org/10.1155/2022/5304860 (https://www.hindawi.com/journals/sci/2022/5304860), https://doi.org/10.1177/2280800018820490 (https://pubmed.ncbi.nlm.nih.gov/30832532/).

-Method and materials are ok

_ New and recent references need to be added in discussion part

-Conclusion section is needed.

Best,

Author Response

Thank you for the comments. Our response is as below:

Reviewer 2

1.       Abstract: aim should be clearly expressed

The aim of the research was expressed In line24-26 which stated that “This research explored the potentials of an inexpensive biogenic HA derived from fish scales and FsHA/YSZ bioceramic coatings on Ti-6Al-4V alloy as an alternative to synthetic HA coatings”.

2.       Introduction needs to be improved and authors should explain the interaction between cell and mentioned materials in their manuscript and they can use the following references:  https://doi.org/10.1155/2022/5304860 (https://www.hindawi.com/journals/sci/2022/5304860), (https://pubmed.ncbi.nlm.nih.gov/30832532/).  

Done. Line 63-68.

3. Method and materials are ok

Thank you.

4. New and recent references need to be added in discussion part

Done. Thank you. 

Reviewer 3 Report

The authors investigated morphology and biomechanical properties of plasma sprayed FsHA/YSZ coatings on Ti-6Al-4V substrate. In the study, an inexpensive biogenic HA derived from fish scales was combined with YSZ to form bioceramic coating powders as an alternative to synthetic HA coatings. The results showed that the powders had slightly irregular morphology and fine spherical morphology while the coatings possessed fine lamellar morphology with partially melted and unmelted FsHA particles and fine micro cracks along with evenly dispersed ZrO2 particles. More interestingly, the coated samples exhibited improvement in surface roughness, hardness, corrosion resistance, and cell viability. The work is interesting and can be published in Coatings if the following issues can be addressed:

1. Some abbreviations should be defined when they are first mentioned in the manuscript (Ex. XRD, SEM, YSZ, SBF…). SBF and SFB were not defined but both were used in the manuscript. The authors should clarify this.

2. Carbon based nanomaterials can precipitate HA crystals and, therefore, enhance bone affinity. The works of https://doi.org/10.1016/B978-0-12-812667-7.00001-X and https://doi.org/10.1016/B978-0-08-102722-6.00006-7 should be cited in the introduction for better review of the applications of carbon-based materials for bone regeneration. 

3.   There is lack of information about the equipment and sources of chemicals used in this study. The authors should provide the information comprehensively in section 2 of the revised manuscript.

4. Figure 1 needs to be improved: scale bars are required and better indications for Figure 1b should be provided to avoid confusion (ex. i), ii) and iii))

5. Why were the samples polished so many times? What was the targeting surface roughness for the polishing steps?

6. Why did Figure 4a have colorful symbols but the others use black ones? The authors should improve the consistency of all Figures in the revised manuscript.

7. Figures 5, 6, 7, 10, 11 are very messy. The indications of the SEM images are too large, creating distraction and covering information that the images could show. The authors should address this issue.

8. How were the regions for EDX analysis chosen? Why were regions with less white areas in Figure 7d not chosen for analysis?

9. In table 4, why were the values of βa, Ecorr, Rp, OCV for FsHA+20wt% YSZ much higher than those of the other samples?

10. The center regions of Figures 11a-c were analyzed by EDX. Why was Figure 11d analyzed at the corner?   

11. Several grammar and typo errors in the manuscript need to be corrected. Writing needs to be improved.

Author Response

Thank you for the comments. Our response is as below:

Reviewer 3

1. Some abbreviations should be defined when they are first mentioned in the manuscript (Ex. XRD, SEM, YSZ, SBF…). SBF and SFB were not defined but both were used in the manuscript. The authors should clarify this.

The abbreviations have been defined where they were first mentioned. SFB was a typo error as the correct one is simulated body fluid (SBF).

2. Carbon based nanomaterials can precipitate HA crystals and, therefore, enhance bone affinity. The works of https://doi.org/10.1016/B978-0-12-812667-7.00001-X and https://doi.org/10.1016/B978-0-08-102722-6.00006-7 should be cited in the introduction for better review of the applications of carbon-based materials for bone regeneration. 

Done. Line 61-64.

3.   There is lack of information about the equipment and sources of chemicals used in this study. The authors should provide the information comprehensively in section 2 of the revised manuscript.

The source of chemicals used has been mentioned. The equipment used in this study were all stated in the manuscript such as the plasma coating machine, surface roughness tester, hardness tester, Gamry instrument. 

4. Figure 1 needs to be improved: scale bars are required and better indications for Figure 1b should be provided to avoid confusion (ex. i), ii) and iii))

The figure has been improved with better indications for the Figure 1b but no scale bar was added as the figure is a picture and not SEM or microscope image.

5. Why were the samples polished so many times? What was the targeting surface roughness for the polishing steps?

The samples were polished in order to obtain a clear image of the microstructure of the coatings by the SEM machine.

6. Why did Figure 4a have colorful symbols but the others use black ones? The authors should improve the consistency of all Figures in the revised manuscript.

The colourful symbols in Figure 4a have been changed.

7. Figures 5, 6, 7, 10, 11 are very messy. The indications of the SEM images are too large, creating distraction and covering information that the images could show. The authors should address this issue.

The figures have been improved.

8. How were the regions for EDX analysis chosen? Why were regions with less white areas in Figure 7d not chosen for analysis?

The regions selected for EDX analysis were randomly selected that was why four regions were selected for each sample to get adequate information on the elemental constituents of each coating.

9. In table 4, why were the values of βa, Ecorr, Rp, OCV for FsHA+20wt% YSZ much higher than those of the other samples?

The values of βa, Ecorr, OCP are not compared literally with other samples, but the corrosion rates (CR) can be compared with one another. In this case, the CR of FsHA+20wt% YSZ is not much lower compared to the other doped samples but much lower than the undoped coating.

10. The center regions of Figures 11a-c were analyzed by EDX. Why was Figure 11d analyzed at the corner?  

Figure 11d was analysed at the corner due to the uniqueness of that region as other regions are similar to the regions analysed in other samples.

11. Several grammar and typo errors in the manuscript need to be corrected. Writing needs to be improved.

Errors have been corrected. Thank you.

Round 2

Reviewer 2 Report

Dear, 

the revised manuscript is acceptable.

Best